# An Insight into Saponins from Quinoa (*Chenopodium quinoa* Willd): A Review

**DOI:** 10.3390/molecules25051059

**Published:** 2020-02-27

**Authors:** Khadija El Hazzam, Jawhar Hafsa, Mansour Sobeh, Manal Mhada, Moha Taourirte, Kamal EL Kacimi, Abdelaziz Yasri

**Affiliations:** 1Laboratory of Natural Resources Valorization, Department of AgroBioSciences, Mohammed VI Polytechnic University Benguerir, Ben Guerir 43150, Morocco; Khadija.ELHAZZAM@um6p.ma (K.E.H.); Jawhar.HAFSA@um6p.ma (J.H.); mansour.sobeh@um6p.ma (M.S.); manal.mhada@um6p.ma (M.M.); 2Laboratory of Bio-Organic and Macromolecular Chemistry, Department Chemical Sciences, Faculty of Science and Technology, Cadi Ayad University, Marrakech 40000, Morocco; taourirte@gmail.com; 3Industrial Executive Operations Division, Gantour Industrial Site, Act 4 Community Gantour, OCP, Youssoufia 46303, Morocco; kamal.elkacimi@ocpgroup.ma

**Keywords:** *Chenopodium quinoa*, Saponins, Saponins elimination methods, Extraction methods, Biological activities

## Abstract

Saponins are an important group found in *Chenopodium quinoa*. They represent an obstacle for the use of quinoa as food for humans and animal feeds because of their bitter taste and toxic effects, which necessitates their elimination. Several saponins elimination methods have been examined to leach the saponins from the quinoa seeds; the wet technique remains the most used at both laboratory and industrial levels. Dry methods (heat treatment, extrusion, roasting, or mechanical abrasion) and genetic methods have also been evaluated. The extraction of quinoa saponins can be carried out by several methods; conventional technologies such as maceration and Soxhlet are the most utilized methods. However, recent research has focused on technologies to improve the efficiency of extraction. At least 40 saponin structures from quinoa have been isolated in the past 30 years, the derived molecular entities essentially being phytolaccagenic, oleanolic and serjanic acids, hederagenin, 3β,23,30 trihydroxy olean-12-en-28-oic acid, 3β-hydroxy-27-oxo-olean-12en-28-oic acid, and 3β,23,30 trihydroxy olean-12-en-28-oic acid. These metabolites exhibit a wide range of biological activities, such as molluscicidal, antifungal, anti-inflammatory, hemolytic, and cytotoxic properties.

## 1. Introduction

The mother grain, quinoa (*Chenopodium quinoa* Willd.), is an annual plant native to the Andes. For over 5000 years, its seeds were the main crop of Andean cultures in South America; however, their cultivation and consumption were virtually eliminated and only remained in farming traditions after the arrival of the Spaniards [1,2].

The genus *Chenopodium* (Chenopodiaceae) comprises 250 species, among them *C. quinoa* [3]. The latter is considered a pseudocereal because its seeds are used as cereal grains [4]. The plants are tall, branched, and have a long vegetative period and broad leaves. They produce large to small flat grains, oval, usually pale yellow, but color can vary from pink to black [5].

Quinoa can be grown in the fertile soils and warm climate of the Andean region of South America, where the crop grows between sea level and around 4000 m above sea level [2,5]. Some quinoa varieties are able to adapt even to marginal environments [6].

The main quinoa-producing countries are Bolivia, Peru, and Ecuador, which produced 146,735 tons in 2017, compared with 59,115 tons in 2007 [7]. In 2017, quinoa production was recorded as 78,657 tons in Peru, 66,792 tons in Bolivia, and 1286 tons in Ecuador [7]. Most quinoa species are still grown in South America, but are also cultivated in the United States, Europe, and Canada. The United States and Europe, especially France, remain large consumers of the crop [4].

Quinoa seeds have remarkable nutritional properties. They contain more proteins, fiber, and fat than regular grains [8]. In particular, their protein content is as high as 15%, with an excellent balance of amino acids, and they are considered an important source of minerals and vitamins [5,9,10]. Due to their high proportion of omega-6 and a significant content of vitamin E, quinoa has been considered as an oilseed crop [4]. The seeds are also rich in neutral lipids. Triglycerides dominate the seeds, comprising about 50%, followed by diglycerides. Linoleic, oleic, and palmitic acids are the main fatty acids found in quinoa seeds [11]. Quinoa flour has a higher content of protein, carbohydrates, and lipids than isolated starch [12]. Polyphenols, phytosterols, and flavonoids have also been documented in the plant, which explains the growing interest it has recently provoked [13,14]. Quinoa has gained recognition for its promise as a valuable functional and medicinal food. The seeds are rich in diverse secondary metabolites with substantial pharmacological activities [15]. On the other hand, many antinutritional substances are found in quinoa, such as saponins, phytic acid, tannins, and trypsin inhibitors which may have a detrimental effect on the growth and performance of monogastric animals using quinoa as their main source of food energy [16,17].

This review provides an updated overview of the saponins of *C. quinoa*, including saponins elimination methods, extraction methods, and different identified structures. It also collects the biological functions of these secondary metabolites as available in the literature.

## 2. Saponins Elimination Methods

Saponins are considered an antinutritional factor, which must be eliminated before the consumption of seeds. To this end, several methods have been reported to wash off seeds’ saponins, including wet methods, dry methods, and combinations of both. Recently, genetic methods have also been developed [5,18].

### 2.1. Wet Methods

Table 1 represents a summary of several wet washing methods. As early as 1978, Rios and his team initially washed quinoa seeds using water at 50 °C [19]. Twelve years later, Ridout et al. (1990) described a method for the separation and analysis of saponins in raw quinoa. They were also the first to classify quinoa according to its saponin content and degree of bitterness. The terms used to describe quinoa were bean, farinaceous, bitter, and astringent, and the rating scale of bitterness was 0: absent, 2: very weak, 4: weak, 6: moderate, 8: strong, and 10: very strong, recorded at times between 0 and 10 min after tasting. Additionally, the authors founded that three sequential washings decreased the total saponin content from 1.03% to 0.18%, a level at which the product lost its unpalatable astringent sensory characteristics. In the same year, Meyer et al. (1990) showed that quinoa seed bitterness is essentially due to the presence of quinoside A [20]. In another experiment, Ruales and Nair (1993) detected two major saponins (1 and 2) in quinoa seeds. After polishing and washing the seeds, 56% of saponin 1 was removed, while saponin 2 was not detectable in the treated samples. Since quinoa did not taste bitter after that washing round, it was assumed that the bitter taste is most likely due to the contribution of saponin 2.

According to Kozioł (1992), the application of the afrosimetric method made it possible to classify the quinoa varieties of the Latinreco experimental plots as “sweet” or “bitter” via taste tests. Koziol suggested that the standard afrosimetric method was developed to allow enough time to extract saponins from quinoa seeds, but proved too long to be used in the field to identify low-saponin varieties or to test the effectiveness of the abrasive peeling. However, with this rapid afrosimetric method, the total time required to perform the analysis was reduced to about 7 min; quinoa is classified as sweet if it gives a foam height of 1.3 cm or less. The application of this method must take into consideration the height of the rapidly changing foam in both directions and the correlation between the height of the foam and the extraction time of the saponins [9].

Unlike the aqueous extraction used in the laboratory, one commercial washing procedure eliminated about 72% of the saponins from the grain, and the saponin content varied from 0% to 2.0% according to the variety—sweet or bitter quinoa [21]. Similarly, Pappier et al. (2008) showed that saponin content decreased to less than 0.06% in seeds treated by industrial process. On the other hand, the saponin content in seeds washed only with water at 50 °C was about 20% of the original content [22]. This suggests that the commercial process was considerably more intense than the laboratory procedure [21].

The initial saponin content in quinoa, 6.34%, reached a level as low as 0.25%–0.01% during the first half hour of washing in a study by Vega-Galvez et al. (2010). Based on these results, the minimum washing time required to extract the largest amount of saponin (represented by a 96% decrease in the saponin content) could be estimated as 60 min [23].

In a different approach, Quispe et al. (2012) described the kinetics of saponin leaching during the washing of quinoa seeds under different temperature conditions using empirical models based on Fick’s diffusional equation for mass transfer to fit the experimental data. At the beginning of the washing process, the leaching of saponins was very rapid with an asymptotic value of saponin concentration in the seeds, and the value was negatively correlated with the washing temperature. The authors were also able to calculate an effective diffusion coefficient by applying the Arrhenius equation, which was found to increase with increasing washing temperature [10]. By the same means, Irigoyen and Giner (2018) used the diffusion equation to analyze the extraction kinetics of saponins, with the aim of perfecting the saponin leaching mechanism. In this study, treatment carried out at 40 °C for 6 min was considered optimum to reach a safe level of saponins for human consumption without visible damage to the seed [24].

From a different perspective, Ruales and Nair (1993) reported the effect of washing on the nutritional quality of the protein content in quinoa seeds, where the digestibility of the quinoa proteins was studied based on pH variation and amino acid composition after digestion with enzymes. The study did not find any change in the amino acid composition after washing, which showed that the process of saponin removal has no effect on the protein nutritional quality of quinoa seeds [25]. In the same regard, Telleria Rios and Sgarbieri (1978) did not find any change in the amino acid composition [19]. However, Nickel et al. (2016) studied the effects of five types of treatment on the saponin content, total phenol content, and antioxidant capacity of quinoa grains grown in Brazil. It was found that washing the seeds with water was not highly efficient in decreasing the saponin content and, consequently, reducing bitterness. In addition, this method resulted in the increase of total phenols [26]. On the other hand, the total phenol content and the antioxidant capacity were found to be higher in water-cooked seeds compared to other treatments. The authors also ran a microscopic examination after washing the grain samples and after cooking to evaluate the damage to the pericarp and seed coat layers. Washed and abraded samples exhibited significant differences when examined under microscope before and after cooking; the abraded samples revealed more cellular content [27]. A study by Pappier et al. (2008) evaluated the effects of saponin removal on fungal contamination. The results of this work showed that all samples containing saponins had 100% fungal contamination. *Penicillium* and *Aspergillus* were the most prevalent contaminants [22].

### 2.2. Dry Methods

Brady et al. (2007) evaluated by HPLC analysis the effects of three types of heat treatment, namely steam pre-conditioning, extrusion, and roasting of quinoa flour on their chemical profile. They showed that the heating temperature caused saponin degradation and therefore could have direct effects on sensory perception and pharmacological properties, and thus the application of extrusion and roasting can reduce the bitter taste conferred by saponins [29]. Gómez-Caravaca et al. (2014) studied the effect of two different degrees of beading (20% and 30%) on the elimination of saponins, and their impact on the phenolic composition of quinoa seeds. Their study proved that, according to the GC-MS analysis results, beading with a degree of abrasion of 30% allowed sweet quinoa to be obtained. The beading technique caused a 21.5% and 35.2% reduction in the free and bound phenolic compounds, and was an efficient method for obtaining sweet quinoa without affecting the total content of phenolic compounds in an excessive way [30]. In another study, Repo-Carrasco-Valencia et al. (2010) explored the effect of roasting on the nutritional value of the seeds of three cereals of Andean origin, among them quinoa (*C. quinoa*), and studied the effects of roasting and boiling on the nutritional value of these seeds, specifically the availability of iron, calcium, and zinc. According to their study, the analysis of zinc dialyzability in different raw samples showed that roasting treatment had no significant effect on the availability of these minerals. In contrast, boiling treatment decrease the dialyzability of iron and zinc in all samples and improved that of calcium [31]. Moreover, the roasting process, according to Nickel et al. (2016), resulted in a reduction of phenolic compounds [26]. In the same regard, the dry method was shown to significantly decrease the vitamin and mineral contents, especially potassium, iron, and manganese [25].

Scarification is the most used dry method at the industry level, and it consists of separating the episperm from the quinoa grain, where the highest content of saponins is concentrated. Studies have shown that the use of a scarificator for 6 min reduces the saponin content in quinoa seeds from 0.324% to 0.001%; however, it damages the seed structure, as small numbers of broken grains were observed, with a minor problem of dust remaining in the scarified grain [32]. Armada (2012) recommended the addition of a polishing operation after scarification, in order to improve the efficiency of the scarification equipment in the quinoa grain disaponification operation [32].

### 2.3. Genetic Methods

Other techniques of removal of quinoa seed saponins have been tested, including genetic methods [5]. A study by Ward (2000) performed selection of quinoa plants to estimate the heritability of saponin content from one generation to the next. This study showed a slow decrease in saponin content; however, the percentage of plants containing less than 1 mg/g of saponin in the grains increased from 3.57% at S1 to 11.1% at S4. This slow selection was explained by the dominance of the allele responsible for saponin expression, which required more time for the recessive inhibitory alleles of saponin synthesis to accumulate at the relevant loci [18]. On the other hand, Dick Mastebroek et al. (2001) selected sweet quinoa genotypes with low concentrations of saponins at different stages of development, based on the concentration of saponins in the leaves. The results showed that the seeds contain higher levels of sapogenins than the leaves, and that the difference in sapogenin content between leaves and seeds is higher in bitter quinoa. Nonetheless, the sapogenin content of the leaves generally increases during the development of the plant, with a maximum content of saponins reached 82 days after sowing. The authors demonstrated that following the sapogenin content in the leaves during the development of the plant did not allow the early selection of sweets genotypes before anthesis [33].

Concentrations of saponins vary between genotypes, from sweeter to bitter varieties. Water and nutrient availability have been found to be responsible for quinoa saponin biosynthesis in seeds, thus strengthening the hypothesis of the combined influence of genetic and environmental factors [33].

In 1989, Risi and Galway evaluated 294 accessions for their ability to adapt to the new environment of England. In this study, the differences between accessions were highly significative for saponin content. This opened the door for other studies to unlock the mystery of saponins in quinoa seeds [34]. In 1996, Jacobsen et al. studied the stability of various descriptive characteristics over a 5 year period in 14 lines of quinoa [35], where saponin content was assessed either by bitterness in a taste test of mature seeds, on a scale from 0–10, or by an afrosimetric method that involved the estimation of foam development [9]. Genotype–environment interactions were significant, with variance components accounting for 16% of the total variation in saponin content. This finding qualifies the saponin content as a quantitative trait. The complexity of understanding saponin expression is mainly due to the cytogenetics of quinoa as an allotetraploid species (2n = 4x = 36, with basic chromosome number of x = 9), with a diploid type of chromosomal segregation [36], although some tetrasomic inheritance occurs in the linkage groups [18]. Although quinoa displays disomic inheritance for most qualitative traits, this influences trait segregation and may cause segregation distortion [37].

Later, a microarray analysis by Reynold et al. (2009) allowed the identification of a set of candidate genes transcriptionally related to saponin biosynthesis, including genes with shared homology to cytochrome P450s, cytochrome P450 monooxygenases, and glycosyltransferases [38].

The development of the quinoa market in non-Andean countries and the development of molecular tools for breeding has inspired researchers to look for a potential new approach to quinoa grain improvement related to saponin content as an economically important trait. Maughan et al. (2004) and Jarvis et al. (2008) published linkage maps based on different molecular resources developed using quinoa RILs (recombinant inbred lines), including SSR (Simple Sequence Repeat), AFLP (Amplified Fragment Length Polymorphism), and RAPD (Random Amplified Polymorphic DNA) markers; 11S seed storage protein loci; the NOR (Nucleolar Organizer Regions); and the morphological betalain color locus [37,39]. The main objective was to provide a key starting point for the genetic dissection of important traits, including saponin content. To increase the precision of the linkage map and to narrow down the exact position of the candidate gene, Maughan et al. (2012) developed an SNP (Single Nucleotide Polymorphisms)-based map using 511 marker loci, which represented an important genomic tool needed in plant breeding programs. The complexity of the polyploidy of quinoa genome and thus the saponin content trait genetics make the inheritance patterns difficult to understand and to apply in agriculture [40].

A new era of research started in 2017 when the genome was sequenced, accelerating the identification of genes responsible for saponin synthesis [41]. The genome sequence facilitated the identification of the transcription factors likely to control the production of saponins, including a mutation that appears to cause alternative splicing, which is a source of a premature stop codon in sweet quinoa. The genes regulating the absence of saponins in sweet quinoa accessions are unknown. To identify these genes, Jarvis et al. (2017) made a linkage mapping and bulk segregant analysis (BSA) using two populations, segregating for the presence of saponins in the seeds. The results showed quantitative and qualitative differences between the saponins identified in bitter and sweet lines. Additionally, the presence and absence of saponins was correlated with the differences in the thickness of the seed coat, with a bitter line having a significantly thicker seed coat than sweet lines [41].

## 3. Extraction Methods of Quinoa Saponins

The extraction of quinoa saponin can be carried out by several methods (Table 2). The most utilized methods are the conventional technologies of maceration, extraction by reflux, and Soxhlet [42]. These methods are based on the solubility of solute in the solvent. Ethanol and methanol are still considered the best solvents for the extraction of saponins because of their high solubilization properties. In a study by Gee et al. (1993), methanol was found to be the most efficient solvent with which to extract saponin. Unlike grain, infant wheat cereals’ aqueous extracts exhibited similar saponin contents to their methanol extracts. In particular, aqueous cereal extracts showed higher oleanolic to hederagenine ratio than quinoa seed aqueous extracts [21]. Additionally, Dini et al. (2001) showed that the aqueous methanolic extract, which is usually removed in the conventional analytical methods, contained some triterpenoid saponins that were not present in the butanol phase. Therefore, the conventional analytical procedure can lead to the loss of components if the water-soluble fraction is neglected [43]. Research is currently focused on microwave extraction, a technology that improves the efficiency of extraction by reducing extraction time and solvent consumption in addition to enhancing the extract quality [44]. In a study by Gianna et al. (2012), the authors evaluated the reliability of extracting saponins from quinoa seeds with solvents by the application of microwaves, and the effect of different parameters such as temperature, seed/solvent ratio, treatment time, and % solvent on saponin removal percentage were evaluated. The numerical analysis of the results showed that the optimal extraction conditions were using ethanol–water and isopropanol–water 20%, with a treatment time of 20 min at a temperature of 90 °C. The extraction of saponins by microwave method makes it possible to reduce the extraction time 20-fold compared to extraction by Soxhlet, and the use of the alcohol as a solvent makes it possible to extract more saponin than the use of water alone [45].

## 4. Chemical Characterization of Quinoa Saponins

Saponins constitute a large group of glycosides found in various plants. They are polar molecules consisting of a steroid or triterpene aglycone with one or more sugar chains and are characterized by their surfactant properties, which differentiate them from other glycosides [56]. According to their structure, saponins are classified into two types. Steroidal saponins dominate in angiosperm monocotyledonous plants (Figure 1a), and triterpene saponins are generally present in angiosperm dicotyledonous plants, the group to which quinoa belongs (Figure 1b) [57].

Saponins are the major secondary metabolites present in quinoa, found mainly in the outer layer of the seeds [25]. Chemically, quinoa saponins are triterpene glycosides composed of a hydrophilic oligosaccharide bound at C-3 and C-28 to a hydrophobic aglycone. Glucose, galactose, arabinose, glucuronic acid, and xylose are the common oligosaccharide sugars. The aglycone is usually derived from oleanolic acid, hederagenin, phytolaccagenic acid, serjanic acid, spergulagenic acid, gypsogenin, 3β-hydroxy-27-oxo-olean-12-en -28-oic acid, and 3β, 23 α, 30 β-trihydroxy-olean-12-en-28-oic acid [43,49,53,55,58,59].

The biosynthesis of quinoa saponins takes place through the mevalonate pathway via farnesyl diphosphate (FPP), a 15 C structure [60]. Pairs of FPP molecules are linked to give squalene (30 C) [61]. The latter is oxidized to oxydosqualene [62], followed by the formation of β-amyrin (the precursor for all quinoa saponins) under the action of β-amyrin synthase. The first aglycone is oleanolic acid, which is then subjected to several modifications (oxidation, glycosylation, and esterification) producing the other seven aglycones [5].

To date, about 40 saponins have been characterized and reported from quinoa seeds (Table 3). Woldemichael and Wink (2001) isolated and characterized 16 saponins from quinoa seeds (*C. quinoa*) by ^1^H, ^13^C NMR spectroscopy, mass spectrometry, and other chemical methods, including five previously isolated major saponins, and the new saponin 3-*O*-β-d-glucopyranosyl-(1,3)-α-l-arabinopyranosyl phytolaccagenic acid [52]. Dini et al. (2001b) isolated six triterpenoid saponins and elucidated their structures by ^1^H NMR analysis. They identified two new compounds, hederagenin 3-*O*-*β*-d-glucopyranosyl-(1,4)-*β*-d-glucopyranosyl-(1,4)-β-d-glucopyranosyl-28-*O*-β-d-glucopyranoside and spergulagenic acid 3-*O*-α-l-arabinopyranosyl-(1,3)-β-d-glucuronopyranosyl-28-*O*-β-d-glucopyranoside [48]. Similarly, Dini et al. (2001a) isolated six triterpenoid saponins from the seeds of quinoa with two new compounds, phytolaccagenic acid 3-*O*-α-l-arabinopyranosyl-(1,3)-β-d-glucuronopyranosyl-28-*O*-β-d-glucopyranoside and phytolaccagenic acid 3-*O*-β-d-glucopyranosyl-(1,3)-β-d-xylopyranosyl-(1,2)-β-d-glucopyranosyl-28-*O*-β-d-glucopyranoside [43]. In another report, Dini et al. (2002) isolated seven triterpenoid saponins from the seeds of “kancolla”, a sweet variety of *C. quinoa*, with one new compound identified by ^13^C NMR spectroscopy as serjanic acid 3-*O*-β-d-glucopyranosyl-(1,3)-α-l-arabinopyranosyl-28-*O*-β-d-glucopyranoside [63]. Along the same lines, Zhu et al. (2002) identified 12 triterpenoid saponins from seeds of *C. quinoa* using ^13^C NMR analysis, with one new compound, 3-*O*-β-d-glucopyranosyl-(1,3)-α-L-arabinopyranosyl-30-*O*-methyl spergulagenate-28-*O*-β-d-glucopyranoside [51]. In one study by Kuljanabhagavad et al. (2008), 20 triterpene saponins were isolated from *C. quinoa.* Compounds’ structures were elucidated by analyzing chemical and spectroscopic data including 1D- and 2D-NMR. Four compounds were isolated and reported for the first time in quinoa, serjanic acid 3-*O*-α-l-arabinopyranosyl-28-*O*-β-d-glucopyranoside, serjanic acid 3-*O*-β-d-glucuronopyranosyl-28-*O*-β-d-glucopyranoside, 3β-*O*-β-d-glucopyranosyl-(1,3)-α-l-arabinopyranosyl-oxy-23-oxo-olean-12-en-28-oic acid β-d-glucopyranoside, and 3β-*O*-β-d-glucopyranosyl-(1,3)-α-l-arabinopyranosyl-oxy-27-oxo-olean-12-en-28-oic acid β-d-glucopyranoside [49]. Additionally, Verza et al. (2012) identified 10 triterpenoid saponins in *C. quinoa* after UPLC/Q-TOF-MS analysis. Among the identified saponins, three were found to be derived from phytolaccagenic acids, two from serjanic acid, two from oleanolic acid, and two from hederagenin, in addition to an undetermined compound [46].

In general, the majority of saponins are isolated from the seeds of quinoa; however, a few studies have reported some saponins isolated from other parts such as leaves, stem, and roots. Dick Mastebroek et al. (2000) reported the presence of saponins in quinoa leaves, but, so far, major saponins have not been isolated from plant parts other than seeds and flowers.

## 5. Biological Activities of Quinoa Saponins

Saponins are characterized by a bitter taste and are considered toxic in high concentrations. They are present throughout the quinoa plant. To date, there have been no studies justifying the role of quinoa saponins and the limiting factors of their production at the level of quinoa, but their presence is generally considered a defense mechanism against the plant’s natural enemies [21,28].

Saponins exhibit several physicochemical and biological properties (Table 4) [64]. These include antioxidant, analgesic, immunostimulant, antimicrobial, antiviral, and cytotoxic activities, and anti-inflammatory and hemolytic effects. They also affect the absorption of some minerals and vitamins and the growth rate of the consuming organisms [60,65]. In order to improve the yield and the biological activities of saponins and their applications in the food, cosmetics, agricultural, and pharmaceutical sectors, several techniques of extraction and identification have been developed [42,44]. Woldemichael and Wink (2001) reported the antifungal activity of a crude saponin extract from *C. quinoa.* The extract inhibited the growth of *C. albicans* at the concentration of 50 µg/mL. However, pure saponins (mono-, bi-, and tridesmosidic saponin) exhibited poor antifungal activity with minimal inhibitory concentrations (MICs) of 100 and 500 µg/mL for mono- and bidesmosidic saponin, respectively; the superior activity of the extract might be attributed to synergistic interactions between the multiple components of the extract, in which they may affect several molecular targets at the same time. However, it has also been reported that the carbohydrate chain in saponins has an effect on membrane permeability [52]. This can be explained in terms of the amphiphilic structure of monodesmosidic saponins [52,66,67]. The lipophilic triterpene aglycone allows the saponins to dive into the microbial membrane, while the carbohydrate chain unit at C3 helps the molecule to binds to glycolipids and extracellular glycoproteins [52]. Quinoa saponins have also shown a biocidal effect against *Pomacea canaliculata*, a snail that attacks rice seeds, with a very positive effect on the germination rate of rice seeds [68,69].

To date, saponins have received much attention due to their wide spectrum of pharmacological and biological proprieties [70,71,72]. Saponins have interesting hemolytic properties against erythrocytes and can interact with lipid and cholesterol membranes as surface active molecules and disrupt these membranes by forming pores that that destabilize the membrane [70,71,72,73,74,75]. According to Gee et al. (1993), quinoa saponins can induce membranolytic activity against the mucosal cells of the small intestine. However, the hemolytic activity of saponins depends on the structure of the saponins in question [21]. Vo et al. (2016) investigated the hemolytic activity of 41 triterpenoid saponins and sapogenins with three different types of structural skeletons, and showed a strong relationship between the chemical structure and the biological activity [75]. In fact, according to this study, almost all of the oleanane sapogenins exhibited stronger hemolytic activity, owing to the presence of a carboxyl group at position 28, an *α*-hydroxyl group at position 16, and a *β*-hydroxyl at position 2. However, a reduction of hemolytic effect was observed with the introduction of methyl hydroxyl at positions 23 or 24 and *α*-OH at position 2 [52,75,76].

Inflammation is the first response to infection or injury via the complex biological response of vascular tissues to harmful stimuli such as pathogens, damaged cells, or irritants [77].

The anti-inflammatory activity of saponins from quinoa seeds has been thoroughly investigated and reported in the literature. In a study by Yao et al. (2014), four quinoa saponin fractions, namely Q30, Q50, Q70, and Q90, extracted from *C. quinoa* were assessed for their anti-inflammatory activity. First, seeds were extracted with methanol by maceration, re-dissolved in water, and partitioned with ethyl acetate and butan-1-ol. After that, butan-1-ol saponins were fractionated in a resin column and eluted with 30% (Q30), 50% (Q50), 70% (Q70), and 90% (Q90) methanol. Results showed that the saponin fractions decreased the response of inflammatory mediators, inhibiting the release of inflammatory cytokines including tumor necrosis factor-α and interleukin-6 in lipopolysaccharide-induced RAW264.7 cells [77]. Verza et al. (2012) evaluated the adjuvant activity of *C. quinoa* saponins towards the humoral and cellular immune responses of mice immunized subcutaneously to ovalbumin (OVA). In the study, two fractions of quinoa saponin were tested (FQ70 and FQ90). Results revealed that the effect of FQ70 was significantly greater than that of FQ90. The quinoa saponins were shown to have a very positive effect on the production of cellular and humoral immune responses to OVA in mice [46]. Moreover, quinoa saponins showed a high capacity to potentiate antibody responses (IgG and IgA) by causing an increase in the permeability of the mucosa, allowing increased absorption of the antigen [78].

Other biological properties, such as the effect of quinoa saponin on adipocyte differentiation, have also been demonstrated. Quinoa saponins were able to decrease cell viability and suppress 3T3-L1 adipocytes during the differentiation process [80]. In addition, bidesmosidic and aglycones saponins from quinoa showed cytotoxic effects through apoptosis induction in colon carcinoma Caco-2 cells [49].

## 6. Conclusions

Altogether, 40 saponin structures with eight different aglycones have been isolated and characterized from quinoa (*C. quinoa*). They exhibit a wide array of biological activities, among them antimicrobial, hemolytic, and anti-inflammatory activities, which make them very beneficial agents, especially in the medical sector. Saponins are also considered an antinutritional factor, necessitating their elimination before the consumption of seeds; to this end, several methods are employed, including wet, dry, and genetic. However, to set up a treatment process for the removal of quinoa saponin on an industrial scale, the best design parameters and the most profitable treatment conditions must be determined. Therefore, additional studies are still needed to reduce energy waste, optimize water flow in a continuous wash process, and improve saponin extraction.

## Figures and Tables

**Figure 1 molecules-25-01059-f001:**
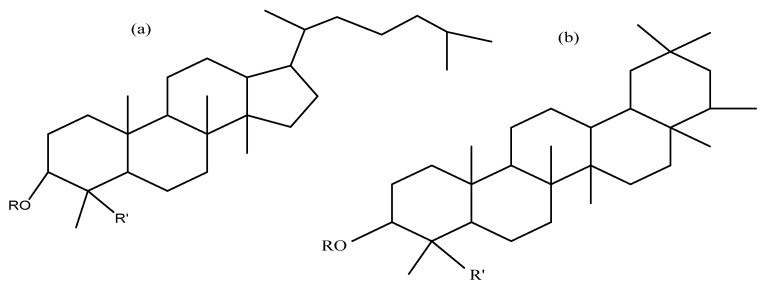
Structures of sapogenins: a steroid (**a**) and a triterpenoid (**b**).

**Table 1 molecules-25-01059-t001:** Wet washing methods of quinoa seeds.

Washing					Drying		% Removal of Saponins	Yield %	Ref.
Number of Wash	Treatment Times (min)	Agitation	Temperature °C	Quinoa: Water	Duration (h)	Temperature °C			
3	30	Yes	-	1:10	16	40	13.44%–84.87%	1.03–0.18	[27]
5	-	No	-	1:10	-	Ambient	72%	0.39	[21]
1	-	Yes	-	1:5	4–5	-	97.7%	0.06	[22]
1	10	Yes	50	1:5	4–5	-	80%	0.16	[22]
1	15–120	Yes	20–60	1:10	7	70	64.35%–80.75%	1.13–0.61	[10]
1	5–70	Yes	20–70	1:5	-	50	80%	0.311–0.062	[24]
-	15		-	-	-	-	17.42%	2.75	[26]
1	0-180	-	-	-	-	40–80	96%	0.25	[23]
1	20	-	-	-	12	50	-	-	[28]

**Table 2 molecules-25-01059-t002:** Summary of saponin extractions methods of *C. quinoa.*

Extraction Technique	Objective	State of Input Material	Solid/Solvent	Solvent Used	Extraction Duration (h)	Extraction Temperature (° C)	Mechanical Aid	Ref.
Maceration	Study of immunoadjuvant activity, toxicity assays, and determination of triterpenic saponins quinoa seeds	Seed powder	1:10	40% EtOH	1	50	Magnetic stirring	[46]
	Study of the anti-inflammatory activity of the saponin extract	Seed powder	1:9	50% EtOH	3		Constant stirring at 200 rpm	[47]
	Evaluation of preconditioning, extrusion, and roasting effects on compounds of quinoa	Seed powder	1:3 1:2	Acetate ethyl MeOH	24 24	-	Magnetic rod	[29]
	Isolation and identification of quinoa saponins	Seed powder	-	90% MeOH	-	-	-	[48]
	Study of saponin extraction kinetics by applying diffusion equations	Seed powder	1:10	50% EtOH	0.5	-	-	[24]
	Isolation and characterization of quinoa saponins	Seed powder	3:25	Petroleum ether MeOH EtOH Bu-OH	-	Room temperature	-	[49]
		Powdered seed coats	3:25					
		Powdered flowers	2:25					
	Isolation and identification of saponins from *C. quinoa*	Seed powder		H_2_O_2_ H_2_O_2_/BuOH	-	60	-	[50]
	Study of the effects of five treatment types on the saponin content, total phenolic compound content, and the antioxidant capacity of quinoa seeds grown in Brazil	Seed powder	1:10	50% EtOH	72	-	-	[26]
	Description of the leaching kinetics of saponins in *C. quinoa*	Seed powder	1:13	50% EtOH	2	-	Constant stirring (200 rpm)	[10]
	Identification of triterpenoid saponins	Seed powder	4kg/-	95% EtOH	3*72	Room temperature	-	[51]
	Study of the influence of drying temperature on kinetic parameters through the application of empirical models	Seed powder	1:10	MeOH	0.5		Magnetic stirrer	[23]
Soxhlet	Isolation and characterization of saponins in quinoa seeds by analytical and chemical methods	Seed powder	900g/-	Petroleum ether MeOH	72	Room temperature	Soxhlet extractor	[52]
	Description of the method for the separation and analysis of saponins in quinoa	Seed powder	1:40	Chloroform MeOH	16 30	-	Soxhlet extractor	[27]
Sonication	Identification of phenolic compounds and saponins of *C. quinoa* Willd	Seed powder	1:15	(4:1) MeOH/Water	1/3	Room temperature	Ultrasonic bath	[53]
	The production of rich saponin extracts from edible seeds (quinoa, soya beans, red lentils, fenugreek, and lupine)	Seed powder	1:10	EtOH EtOH/Water Water	0.25	Room temperature	Ultrasonic probe	[54]
	Optimization of processes of obtaining saponin-rich extracts from different plants	Seed powder	1:10	MeOH	0.25	Room temperature	Ultrasonic probe	[55]
Microwave	Evaluation of the efficiency of saponin extraction	Seed powder	1:20	20% EtOH 20% Isopropanol	1/3	90° C	Microwave	[45]

**Table 3 molecules-25-01059-t003:** Triterpene saponins isolated from *C. quinoa* Willd.

Compound	R1	R2	Formula	MW	Quinoa Part	Origin	Ref.
**Oleanolic Acid** 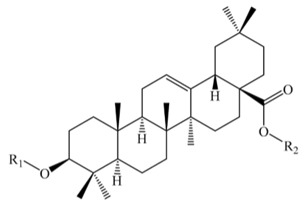							
Monodesmosidic saponins							
Oleanolic acid 3-*O*-β -d-glucopyranoside	-glc	H	C_36_H_58_O_8_	618	Seeds	m	[50]
Oleanolic acid 3-*O*-β-d-glucuropyranoside	-glcUA	H	C_36_H_56_O_9_	632	Seeds	b	[51,52]
Oleanolic acid 3-*O*-β -d-xylopyranosyl-(1,3)-β-d-glucuronopyranoside	–xyl (1–3) glcUA	H	C_41_H_64_O_13_		Seeds	m	[50]
Oleanolic acid 3-*O*-β -d-xylopyranosyl(l,3)-β -d-glucuronopyranosyl-methyl-ester	–xyl (1–3)-6-O-Me glcUA	H	C_42_H_66_O_13_	778	Seeds	m	[50]
Bidesmosidic saponins							
Oleanolic acid 3-*O*- [β -d- glucopyranosyl-(1,3)- α-l- arabinopyranosyl]-28-*O*-β-d-glucopyranoside	–glc (1–3) ara	-glc	C_47_H_76_O_17_	912	Seeds, bran flowers, and fruits	a, b, c, e, g	[43,49,59,63]
Oleanolic acid 3-*O*-α-l-arabinopyranosyl-(1,3)-β-d-glucuronopyranosyl-28-*O*-β-d-glucopyranoside	–ara (1–3) glcUA	-glc	C_47_H_74_O_18_	926	Seeds	a, e, p	[43,59,63]
Oleanolic acid 3-*O*-[β-d-glucuronopyranosyl]-28-*O*-β-d-glucopyranoside	–glcUA	-glc	C_42_H_66_O_14_	794	Flowers, fruits, seeds, and bran	a, c, p, g	[49,52,53,58,59,63]
Oleanolic acid 3-*O*-β-d-xylopyranosyl-(1,3)-β-d-glucuronopyranosyl-28-*O*-β-d-glucopyranoside	–xyl (1–3) glcUA	-glc	C_47_H_74_O_18_	926	Flowers, fruits, seeds, and bran	a, c, g, sa, p	[49,50,53,58,59]
Oleanolic acid 3-*O*-β-d-glucopyranosyl-(1,2)-β-d-glucopyranosyl-(1,3)-α-l-arabinopyranosyl 28-*O*-β-d-glucopyranoside	–glc (1–2) glc (1–3) ara	-glc	C_53_H_86_O_22_	1074	Flowers, fruits, seeds, and bran	a, b, c, g, sa	[49,51,58,59]
Hederagenin 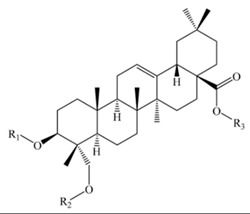							
Monodesmosidic saponins R3= -H							
Hederagenin 3-*O*-β-d-glucopyranosyl-(1,3)- α -l-arabinopyranoside	–glc (1–3) ara	-H	C_41_H_66_O_13_	766	Seeds and bran	b, c	[51,52]
Hederagenin 3-*O*-α-l-Arabinopyranoside	-ara	-H	-	-	Seeds	c	[52]
Bidesmosidic saponins R3= -H							
Hederagenin 3-*O*-[β-d-glucopyranosyl-(1,3)-α-l-arabinopyranosyl]-28-*O*-β-d-glucopyranoside	–glc (1–3) ara	-glc	C_47_H_76_O_18_	928	Flowers, fruits, seeds and bran	a, b, c, e, g, sa, p	[25,48,49,51,52,53,58,59,63]
Hederagenin 3-*O*- α- l-arabinopyranosyl-28-*O*-β-d-glucopyranoside	-ara	-glc	C_41_H_66_O_13_	766	Flowers, fruits, seeds, and bran	a, c, g, sa	[49,52,58,59]
Hederagenin 3-*O*-β-d-Glucopyranosyl-(1,3)-α-l-galactopyranosyl 28-*O*- β-d-glucopyranoside	–glc (1–3) gal	-glc	C_48_H_78_O_19_	958	Flowers, fruits, seeds, and bran	a, c, b, g, sa	[49,51,58,59]
Hederagenin 3-*O*- β- d-Glucuronopyranosyl 28-*O*- β-d-glucopyranoside	–glcUA	-glc	C_42_H_66_O_15_	810	Flowers, fruits, seeds, and bran	a, c, g, sa	[49,58,59]
Hederagenin 3-O-β-d-Xylopyranosyl-(1,3)-β-d-glucuronopyranosyl-28-*O*-β-d-glucopyranoside	–xyl (1–3) glcUA	-glc	C_47_H_74_O_19_	942	Seeds and bran	a, sa	[53,58,59]
Hederagenin 3-*O*-β-d-Glucopyranosyl-(1,4)-β-d-glucopyranosyl-(1,4)-β-d-glucopyranosyl-28-*O*-β-d-glucopyranoside	–glc (1–4) glc (1–4) glc	-glc	C_54_H_88_O_24_	1120	Seeds	a, e	[48,59]
Tridesmosidic saponins R3= -glc							
3,23-bis(*O*-β-d-Glucopyranosyloxy) olean-12-en28-oicacid28-*O*-α-l-arabinopyanosyl-(1,3)-β-d-glucopyranoside	-glc	-glc-glc	C_53_H_86_O_23_	1090	Seeds	e	[20]
Phytolaccagenic acid 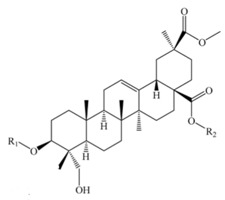							
Monodesmosidic saponins							
Phytolaccagenic acid 3-*O*-β-d-glucopyranosyl (1,3)-α-l-arabinopyranoside	–glc (1–3) ara	-H	C_42_H_66_O_15_	810	Seeds	b, c	[51,52]
Bidesmosidic saponins							
Phytolaccagenic acid 3-*O*- [α -l-arabinopyranosyl-(1,3)-β-d-glucuronopyranosyl]-28-*O*-β-d-glucopyranoside	–ara (1–3) glcUA	-glc	C_48_H_74_O_21_	986	Seeds	e	[43,48]
Phytolaccagenic acid 3-*O*-[β-d-glucopyranosyl-(1,3)-α-l-arabinopyranosyl]-28-*O*-β-d-glucopyranoside	–glc (1–3) ara	-glc	C_48_H_76_O_20_	972	Flowers, fruits, seeds, and bran	a, b, c, e, g, m, p, sa	[43,49,51,52,53,58,59,63]
Phytolaccagenic acid 3-*O*-[β-d-glucopyranosyl-(1,3)-β-d-xylopyranosyl-(1,2)- β-d-glucopyranosyl]-28-*O*-β-d-glucopyranoside	–glc (1–2) xyl (1–3) glc	-glc	C_54_H_86_O_25_	1134	Seeds	a, e, sa	[43,59]
Phytolaccagenic acid 3-*O*-α -l-arabinopyranosyl 28-*O*-β-d-glucopyranoside	-ara	-glc	C_42_H_66_O_15_	810	Flowers, fruits, seeds, and bran	a, b, c, g, p, sa	[49,51,52,53,58,59]
Phytolaccagenic acid 3-*O*-β-d-galactopyranosyl-(1,3)-β-d-glucopyranosyl 28-*O*-β-d-glucopyranoside	–gal (1–3) glc	-glc	C_49_H_78_O_21_	1002	Seeds	b	[51]
Phytolaccagenic acid 3-*O*-β-d-Glucopyranosyl-(1,3)-β-d-galactopyranosyl 28-*O*-β-d-glucopyranoside	–glc (1–3) gal	-glc	C_49_H_78_O_21_	1002	Flowers, fruits, seeds, and bran	a, c, g, sa	[49,53,58,59]
Phytolaccagenic acid 3-*O*-β-d-glucopyranosyl-(1,2)-β-d-glucopyranosyl-(1,3)-α-l-arabinopyranosyl 28-*O*-β-d-glucopyranoside	–glc (1–2) glc (1–3) ara	-glc	C_54_H_86_O_25_	1134	Flowers, fruits, seeds, and bran	a, c, e, g, sa	[43,49,51,53,58,59]
Phytolaccagenic acid 3-*O*-β-d-Glucopyranosyl-(1,4)-β-d-glucopyranosyl-(1,4)-β-d-glucopyranosyl 28-*O*-β-d-glucopyranoside	–glc (1–4) glc (1–4) glc	-glc	C_55_H_88_O_26_	1164	Flowers, fruits, seeds, and bran	a, c, e, g	[48,49,59]
Serjanic acid 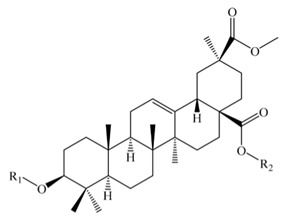							
Bidesmosidic saponins							
Serjanic acid 3-*O*-[*β*-d-glucopyranosyl-(1,3)- α -l-arabinopyranosyl]-28-*O*-*β*-d-glucopyranoside = 30-*O*-methyl spergulagenate 3-*O*-β-d-glucopyranosyl-(1,3)-α-l-arabinopyranosyl-28-*O*-β-d-glucopyranoside	–glc (1–3) ara	-glc	C_48_H_76_O_19_	956	Flowers, fruits, seeds, and bran	a, b, c, g, p	[49,51,53,59,63]
Serjanic acid 3-*O*-β-d-glucopyranosyl-(1,2)-β-d-glucopyranosyl-(1,3)-α-l-arabinopyranosyl-28-*O*-β-d-glucopyranoside= 30-*O*-methyl spergulagenate 3-*O*-β-d-glucopyranosyl-(1,2)-β-d-glucopyranosyl-(1,3)-α-L-arabinopyranosyl-28-*O*-β-d -glucopyranoside	–glc (1–2) glc (1–3) ara	-glc	C_54_H_86_O_24_	1118	Flowers, fruits, seeds, and bran	b, c, e, g, sa	[49,51,58]
Serjanic acid 3-*O*-α-l-arabinopyranosyl-28-*O*-β-d-glucopyranoside	-ara	-glc	C_42_H_66_O_14_	794	Flowers, fruits, seeds, and bran	c, g	[49]
Serjanic acid 3-*O*-β-d-glucuronopyranosyl-28-*O*-β-d-glucopyranoside	–glcUA	-glc	C_43_H_66_O_16_	838	Flowers, fruits, seeds, and bran	c, g	[49]
Serjanic acid 3-*O*-α-l-arabinopyranosyl-(1,3)-β-d-glucuronopyranosyl-28-*O*-β-d-glucopyranosyl ester	-ara (1-3) glc	-glc	C_48_H_74_O_20_	970	Seeds	p	[53]
Spergulagenic acid 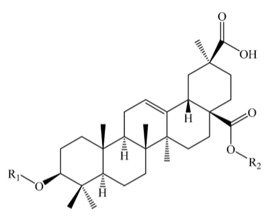							
Bidesmosidic saponins							
Spergulagenic acid 3-*O*-β-d-glucopyranosyl-(1,2)-β-d-glucopyranosyl-(1,3)-α-l-arabinopyranoside	-glc (1-2) glc (1-3) ara	-H	-	-	Bran	sa	[58]
Spergulagenic acid 3-*O*-β-d-glucopyranosyl-(1,2)-β-d-glucopyranosyl-(1,3)-α-l-arabinopyranosyl-28-*O*-β-d-glucopyranoside	Glc (1-2) glc (1-3) ara	-glc	-	-	Seeds	e	[48]
Spergulagenic acid 3-*O*-α-l-arabinopyranosyl-(1,3)-β-d-glucuronopyranosyl-28-*O*-β-d-glucopyranoside	–ara (1–3) glcUA	-glc	C_48_H_74_O_20_	970	Seeds	e	[48]
28-oic acid							
3β-Hydroxy-23-oxo-olean-12-en -28-oic acid= Gypsogenin 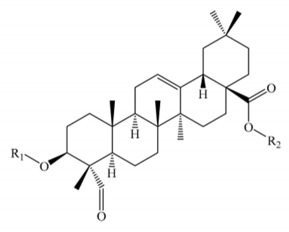							
3β-*O*-β-d-glucopyranosyl-(1,3)- α -l-arabinopyranosyl-oxy-23-oxo-olean-12-en-28-oic acid β-d-glucopyranoside	–glc (1–3) ara	-glc	C_47_H_74_O_18_	926	Flowers, fruits, seeds, and bran	c, g	[49]
3β-Hydroxy-27-oxo-olean-12-en -28-oic acid 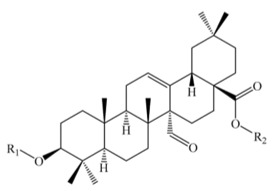							
3β-*O*-β-d-glucopyranosyl-(1,3)-α- l -arabinopyranosyl-oxy-27-oxo-olean-12-en-28-oic acid β-d-glucopyranoside	–glc (1–3) ara	-glc	C_47_H_74_O_18_	926	Flowers, fruits, seeds, and bran	c, g	[49]
3β, 23 α, 30 β-Trihydroxy-olean-12-en-28-oic acid 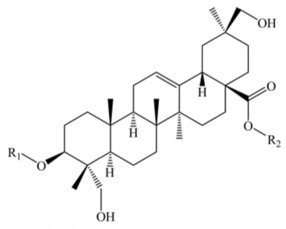							
3,23,30-trihydroxyolean-12-en-28-oic acid 3-*O*-β-d-Glucopyranosyl-(1,3)-α-l-arabinopyranosyl-28-*O*-β-d-glucopyranoside	–glc (1–3) ara	-glc	C_47_H_76_O_19_	944	Flowers, fruits, seeds, and bran	a, c, g	[49,53,59]

**Table 4 molecules-25-01059-t004:** Biological activities of quinoa saponins.

Aglycone	Structure Name	Quinoa Part	Biological Activity	Species	Ref.
-	-	Seeds	Intestinal permeability	Rat	[21]
-	-	Seeds	Molluscicidal activity	*Pomacea canaliculata*	[68,69]
-	-	Seeds	Antifungal activity	*Pyrenophora triticirepentis* and *Rhynchosporium secalis*	[79]
-	-	Seeds		*Botrytis cinerea*	[66]
Phytolaccagenic acid	3-*O*-β-d-glucopyranosyl-(1,3) -α-l-arabinopyranosyl phytolaccagenic acid	Seeds		*Candida albicans*	[52]
	-	Bran	Anti-inflammatory activity	-	[47]
Oleanolic acid	Methyl oleanate	Bran			
	3-*O*-β-d-glucuronopyranosyl oleanolic acid 28-*O*-β-d- glucopyranosyl ester		Hemolytic activity	-	[49,52,58,63]
Serjanic acid	3-*O*-α-l-arabinopyranosyl serjanic acid 28-*O*-β-d-glucopyranosyl ester3-*O*-β-d-glucuronopyranosyl serjanic acid 28-*O*-β-d- glucopyranosyl ester	Seeds, bran, flowers, and fruits	Cytotoxic activity in HeLa cell lines	-	[49,58]
Gypsogenin (3β-Hydroxy-23-oxo-olean-12-en-28-oic acid)	3-*O*-β-d-glucopyranosyl-(1,3)-α-l-arabinopyranosyl 23-oxo-olean-12-en-28-oic acid 28-*O*-β-d-glucopyranosyl ester 3β-[(*O*-β-d-glucopyranosyl-(1,3)-α-l-arabinopyranosyl) oxy]-23-oxo-olean-12-en-28-oic acid 28-*O*-β-d-glucopyranosyl ester				[49]

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
