# Peer review of "An Insight into Saponins from Quinoa (*Chenopodium quinoa* Willd): A Review"

_molecules, 2020, doi:10.3390/molecules25051059_

Round 1

Reviewer 1 Report

This paper described the investigation of saponins from Chenopodium quinoa as a review article. The authors show the isolation, composition, and chemical structure of saponins from Chenopodium quinoa with biological activities. The reviewer feels interesting paper and can agree publication for “Molecules” journal. However The reviewer want to show several comments following. So please answer along my comments.

This paper has no chemical synthesis. Natural product synthesis of saponins are also important manner. The authors have to touch this fields. How is the biosynthetic pathway of saponins? Now, chemical biology containing the genetically research is active. If you have any information, please show a couple of stories. Did the authors submit editing service? There are many mistakes grammar, spell, and rule of this journal, especially “Reference” section. Please check your manuscript deep carefully.

Author Response

  • This paper has no chemical synthesis. Natural product synthesis of saponins are also important manner. The authors have to touch this fields. How is the biosynthetic pathway of saponins?

# Several documents [[1–5] have already discussed this approach in which the authors have detailed the pathway for biosynthesis of saponins regardless of quinoa saponin or any other plant. But considering the importance of this approach, we have added a summary of the biosynthesis of saponin you will find it in section 4. Chemical characterization of quinoa saponins between lines 259 and 263.

The biosynthesis of quinoa saponins takes place through the mevalonate pathway via farnesyl diphosphate (FPP), a 15 C structure [60]. Each 2 FPP molecules are linked to give squalene (30 C) [61]. The latter is oxidized to oxydosqualene [62], followed by the formation of β-amyrin (the precursor for all quinoa saponins) under the action of β-amyrin synthase. The first aglycone is oleanolic acid which is then subjected to several modifications (oxidation, glycosylation, and esterification) producing the other 7 aglycones [5].

References

  1. Bhargava, A.; Srivastava, S. Quinoa: botany, production and uses; CABI: Wallingford, Oxfordshire, UK ; Boston, MA, 2013; ISBN 978-1-78064-226-0.
  2. Kuljanabhagavad, T.; Thongphasuk, P.; Chamulitrat, W.; Wink, M. Triterpene saponins from Chenopodium quinoa Willd. Phytochemistry 2008, 69, 1919–1926.
  3. Osbourn, A.; Goss, R.J.M.; Field, R.A. The saponins – polar isoprenoids with important and diverse biological activities. The Royal Society of Chemistry 2011, 28, 1261.
  4. Holstein, S.; Hohl, R. Isoprenoids: Remarkable diversity of form and function. Lipids 2004, 39, 293–309.
  5. Haralampidis, K.; Trojanowska, M.; Osbourn, A. Biosynthesis of triterpenoid saponins in plants. Adv Biochem Eng Biotechnol 2001, 75, 31–49.
  • Now, chemical biology containing the genetically research is active. If you have any information, please show a couple of stories.

#Concerning the genetic aspect of saponins. The literature is oriented mainly towards the genetic selection of the sweet varieties and the production of mutations in the responsible gene in order to eliminate the saponin, we have added some stories in the genetic methods part between lines 178 and 218, highlighted.

  • Did the authors submit editing service? There are many mistakes grammar, spell, and rule of this journal, especially “Reference” section. Please check your manuscript deep carefully.

#We corrected them.

Reviewer 2 Report

The manuscript "An insight into saponins from Chenopodium quinoa: a Review" by Khadija El Hazzam, Jawhar Hafsa, Mansour Sobeh, Manal Mhada, Moha Taourirte, Kamal EL Kacimi and Abdelaziz Yasri (Molecules-710321) reviews recent progress in isolation, identification and removal of saponins from the seeds of Quinoa. The manuscript is generally well organized and the subject deserves the review.

I have two minor comments to consider:

The conclusions are written in a rather inconsistent way. Especially the following fragment is incomprehensible: “In addition, leaching of minerals will be measured to know the extent of this possible phenomenon. Such a model, combined with mass balances that include the water increase content, will be useful for equipment design, by predicting saponins extraction times for different operating conditions”. Honestly, it looks a bit like a fragment of conclusions from some other manuscript... A phrase starting at the bottom of page 16 is not clear: “In the same regard, mono-, bi- and tridesmosidic saponin have poor antifungal activity, which indicates that the carbohydrate chain in saponins is critical for the membrane permeability and hence, the growth inhibition of fungi”. If indeed all types of saponins (mono-, bis- and tridesmosidic) have poor activity – how can we say anything about the role of the carbohydrate chain ?

Overall, I find the manuscript interesting but would suggest revision prior to its acceptance.

Author Response

  • The conclusions are written in a rather inconsistent way. Especially the following fragment is incomprehensible: “In addition, leaching of minerals will be measured to know the extent of this possible phenomenon. Such a model, combined with mass balances that include the water increase content, will be useful for equipment design, by predicting saponins extraction times for different operating conditions”. Honestly, it looks a bit like a fragment of conclusions from some other manuscript.

#We improved it.

  • A phrase starting at the bottom of page 16 is not clear: “In the same regard, mono-, bi- and tridesmosidic saponin have poor antifungal activity, which indicates that the carbohydrate chain in saponins is critical for the membrane permeability and hence, the growth inhibition of fungi”. If indeed all types of saponins (mono-, bis- and tridesmosidic) have poor activity – how can we say anything about the role of the carbohydrate chain?

# In literature monodesmosidic have better antifungal activity than bi and tridesmosidic saponin, which indicates that the carbohydrate chain in saponins has an effect on membrane permeability [6,7]. However, we corrected it.

References

  1. Woldemichael, G.M.; Wink, M. Identification and Biological Activities of Triterpenoid Saponins from Chenopodium quinoa. Journal of Agricultural and Food Chemistry 2001, 49, 2327–2332.
  2. ZOUAOUI, S.A.; MEGHERBI-BENALI, A.; TOUMI Benali, F.; OUAAR, D. Contribution to the study of the antifungal potency of the seeds of Chenopodium quinoa wild against two phytopathogenic fungi of the barley: Pyrenophora tritici-repentis and Rhynchosporium secalis. Bulletin de la Société Royale des Sciences de Liège 2018, 87, 100–111.

Overall, I find the manuscript interesting but would suggest revision prior to its acceptance.

Round 2

Reviewer 1 Report

Dear author

It is OK to publish the journal.

regards